# $R^2$-VOS: Robust Referring Video Object Segmentation via Relational Cycle Consistency

## Abstract

Referring video object segmentation (R-VOS) aims to segment the object masks in a video given a referring linguistic expression to the object. It is a recently introduced task attracting growing research attention. However, all existing works make a strong assumption: The object depicted by the expression must exist in the video, namely, the expression and video must have an object-level semantic consensus. This is often violated in real-world applications where an expression can be queried to false videos, and existing methods always fail in such false queries due to abusing the assumption. In this work, we emphasize that studying semantic consensus is necessary to improve the robustness of R-VOS. Accordingly, we pose an extended task from R-VOS without the semantic consensus assumption, named Robust R-VOS ($R^2$-VOS). The $R^2$-VOS task is essentially related to the joint modeling of the primary R-VOS task and its dual problem (text reconstruction). We embrace the observation that the embedding spaces have relational consistency through the cycle of text-video-text transformation, which connects the primary and dual problems. We leverage the cycle consistency to discriminate the semantic consensus, thus advancing the primary task. Parallel optimization of the primary and dual problems are enabled by introducing an early grounding medium. A new evaluation dataset, $R^2$-Youtube-VOS, is collected to measure the robustness of R-VOS models against unpaired videos and expressions. Extensive experiments demonstrate that our method not only identifies negative pairs of unrelated expressions and videos, but also improves the segmentation accuracy for positive pairs with a superior disambiguating ability. Our model achieves the state-of-the-art performance on Ref-DAVIS17, Ref-Youtube-VOS, and the novel $R^2$-Youtube-VOS dataset.

## 1 Introduction

Referring video object segmentation (R-VOS) aims to segment a referred object in a video sequence given a linguistic expression. R-VOS has witnessed growing interest thanks to its promising potential in human-computer interaction applications such as video editing and augmented reality. Unlike other video segmentation tasks [45, 36, 35, 46] that only rely on visual cues, R-VOS [13] pairs a target video with a linguistic expression referring to an object.

Previous works [1, 44] tackle the R-VOS problem with a strong assumption that the referred object exists in the video, i.e., there is an object-level semantic consensus between the expression and the video. However, this assumption does not always hold in practice. As shown in Figure 1, we notice a severe false-alarm problem experienced by previous methods when the semantic consensus does not exist, which may prevent those methods from being useful in various applications that cannot provide accurate vision-language pairs. We argue that the current R-VOS task is not completely defined with the assumption that the referred object always exists in the video.

Even when semantic consensus exists in the given video-language pairs, it is still challenging to locate the correct object in the video due to the multimodal nature of the R-VOS task. Recently, MTTR [1]

Submitted to 36th Conference on Neural Information Processing Systems (NeurIPS 2022). Do not distribute.

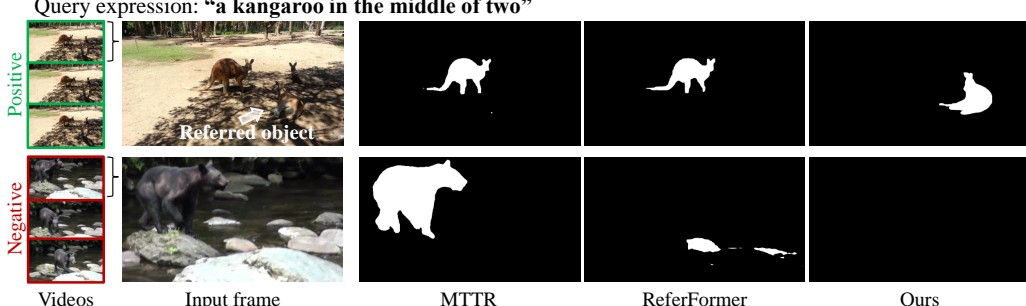

Figure 1: Illustration of the new $R^2$-VOS task. A linguistic expression is given to query a set of videos without the semantic consensus assumption. Videos containing the referred object by the expression are positive, otherwise negative. Unlike the previous R-VOS setting that assumes all target videos are positive to the query expression, the new $R^2$-VOS task is required to discriminate positive and negative text-video pairs, and further segment object masks for all frames in positive videos or treat entire negative videos as backgrounds. Compared to the previous state-of-the-art R-VOS methods, MTTR [1] and ReferFormer [44], our method not only discriminates negative videos better but also shows a superior disambiguating ability between visually similar objects in positive videos.

employs a multimodal transformer encoder to learn a joint representation of the linguistic expression and video, and then obtains the referred object by ranking all objects in the video. ReferFormer [44] follows the image-level method, ReTR [19], to adopt the linguistic expression as a query to the transformer decoder to avoid redundant ranking of all objects. However, these latest methods suffer from semantic misalignment of the segmented object and the linguistic expression, even with sophisticated components employed. As shown in Figure 1, the segmented objects by MTTR and ReferFormer are often not the object referred to by the linguistic expression.

In this paper, we seek to investigate the semantic alignment problem between visual and linguistic modalities in referring video segmentation. We extend the current task definition of R-VOS [13] to accept both paired and unpaired video and language inputs. This new task, which we term Robust R-VOS ($R^2$-VOS), overcomes the current limitation of the R-VOS task by additionally considering the semantic alignment of input video to referring expression. We reveal that this task is essentially related to two problems that are interrelated [31]: the R-VOS problem as the **primary** problem of segmenting mask sequences from videos with referring texts, and its **dual** problem of reconstructing text expressions from videos with object masks. By linking the primary and dual problems, we introduce a text-video-text cycle and a corresponding relational consistency constraint, which can enforce the semantic consensus between the text query and segmented mask to improve the primary task. In practice, naively conducting cyclic training of the text-video-text cycle will lead to a two-stage regime and significantly increasing costs. We address this problem by incorporating an early grounding scheme, serving as a proxy, to efficiently model the two tasks in a parallel manner. In addition, we discriminate the semantic misalignment between the video and text by assessing the cycle consistency between the original and reconstructed texts, thus alleviating the false-alarm problem. Our contributions can be summarized as:

- We notice a severe false-alarm problem faced by previous R-VOS methods with unpaired inputs. To investigate the robustness of current referring segmentation models, we introduce the $R^2$-VOS task that accepts unpaired video and text as inputs.

- We propose a pipeline that jointly optimizes the primary referring segmentation and dual expression reconstruction task and introduces a relational cycle consistency constraint to enhance the semantic alignment between visual and textual modalities.

- Our method surpasses previous state-of-the-art methods on Ref-Youtube-VOS, Ref-DAVIS, and $R^2$-Youtube-VOS dataset in terms of both performance and speed.

## 2 Related Works

**Vision and language representation learning.** There have been a long line of studies on how to learn better vision-language representation, e.g., multimodal attention [30, 50, 8, 3], fusion scheme [7, 14, 15, 51], multi-step reasoning [47, 10] and pretraining [37, 5, 17]. KAC Net [2] leverages

knowledge-aided consistency constraints to enhance semantic alignment for weakly supervised phrase grounding. A structure-preserving constraint [42] is proposed to preserve some intra-modal properties when learning vision-language representation for image-text retrieval.

**Referring video object segmentation.** R-VOS is a novel task that aims to segment an object across frames given a linguistic description. URVOS [39] is the first unified R-VOS framework with a cross-modal attention and a memory attention module, which largely improves R-VOS performance. ClawCraneNet [21] leverages cross-modal attention to bridge the semantic correlation between textual and visual modalities. ReferFormer [44] and MTTR [1] are two latest works that utilize transformers to decode or fuse multimodal features. ReferFormer [44] employs a linguistic prior to the transformer decoder to focus on the referred object. MTTR [1] leverages a multimodal transformer encoder to fuse linguistic and visual features. Different from other vision-language tasks, e.g., image-text retrieval [25, 26, 32] and video question answering [18, 40], R-VOS needs to construct object-level multimodal semantic consensus in a dense visual representation.

# 3 $R^2$-VOS

## 3.1 Task Definition

We introduce a novel task, robust referring video segmentation ($R^2$-VOS), which aims to predict mask sequences $\{M_o\}$ for an unconstrained video set $\{V\}$ given a language expression $E_o$ of an object $o$. Different from the previous R-VOS setup, the queried video $V$ is not required to contain the referred object by expression $E_o$. We define a video $V$ and an expression $E_o$ to have **semantic consensus** if the object $o$ appears in $V$, and the video is **positive** with respect to $E_o$, otherwise it is **negative**. The $R^2$-VOS task is extended to discriminate positive and negative videos, and predict masks $M_o$ of object $o$ for positive videos and treat all frames in the negative videos as background.

## 3.2 Problem Analysis

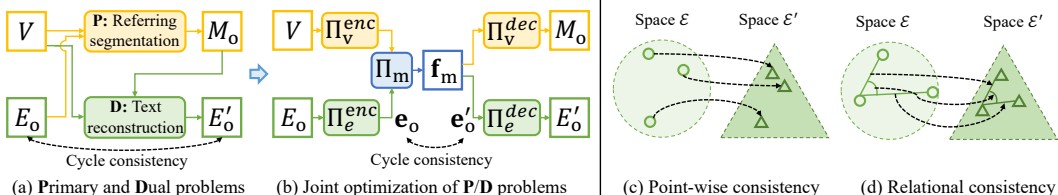

| (a) **P**rimary and **D**ual problems | (b) Joint optimization of **P/D** problems | (c) Point-wise consistency | (d) Relational consistency |

Figure 2: Problem analysis. (a) $R^2$-VOS introduces the **P**rimary problem of referring segmentation and the **D**ual problem of text reconstruction for positive videos. The **P/D** problems are connected in a cycle path from original expression $E_o$ to reconstructed expression $E_o'$. (b) The cycle consistency between the original and reconstructed embeddings ($\mathbf{e}_o$ and $\mathbf{e}_o'$) can benefit to optimize the **P** problem. We enable the joint optimization for cycle consistency with a cross-modal proxy $\mathbf{f}_m$ defined between all single-modal operations (i.e., $\Pi_v^{enc}$, $\Pi_e^{enc}$, $\Pi_v^{dec}$ and $\Pi_e^{dec}$). (c) Point-wise consistency is not suitable in $R^2$-VOS because the mapping between $\mathcal{E}$ and $\mathcal{E}'$ are not necessarily bijective. An object can be referred by various textual expressions. (d) Instead, we apply a relational consistency to preserve distances and angles.

**Primary and dual problems for $R^2$-VOS.** The referring segmentation can be formulated as the maximum *a posteriori* estimation problem of $p(M_o|V, E_o)$. By applying the Bayes rule, we obtain:

$$p(M_o|V, E_o) \sim p(E_o|V, M_o)p(M_o|V) \tag{1}$$

As the prior $p(M_o|V)$ is not affected by the expression $E_o$, we consider maximizing $p(E_o|V, M_o)$ as a dual problem of the referring segmentation (primary problem), which is to reconstruct the text expression given the video and object masks. We note that for negative videos, $p(E_o|V, M_o)$ is undefined because the mask $M_o$ is empty. Thus, we only investigate the dual problem for positive videos. The primary problem and the dual problem can be connected in a cycle path, i.e., from the original expression $E_o$ to the reconstructed expression $E_o'$ through positive video queries, as shown in Figure 2 (a). We believe that the cycle constraint benefits to optimize the primary problem by enhancing the semantic consensus.

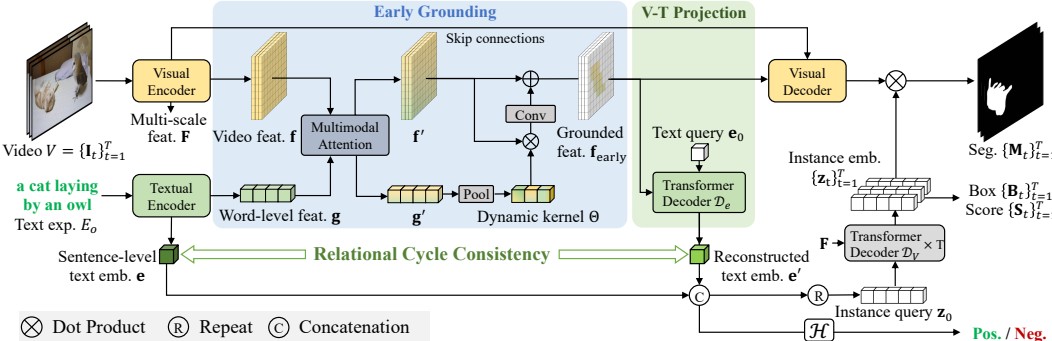

Figure 3: Overview of the proposed model. Given a video clip $V = \{\mathbf{I}_t\}_{t=1}^{T}$ and a textual expression $E_o$ referring object $o$, we first extract video feature and text feature separately, then fuse them in the early grounding module to obtain the visual representation $\mathbf{f_{early}}$ of the referred object $o$. Then we project $\mathbf{f_{early}}$ to a textual space to be $\mathbf{e}'$ and add the relational cycle constraint with the original text embedding $\mathbf{e}$. The final segmentation is obtained by dynamic convolutions with video features from the visual decoder and dynamic weights from the fused text embeddings. The semantic consensus of input pairs is discriminated to be positive or negative by assessing the consistency between $\mathbf{e}$ and $\mathbf{e}'$.

In practice, we study the cycle consistency between the original textual embedding space $\mathcal{E}$ and the transformed textual embedding space $\mathcal{E}'$ induced by positive videos. By definition, the path from the original text embedding $\mathbf{e}_o$ to the reconstructed text embedding $\mathbf{e}'_o$ is modulated with **cross-modal** interactions between video and text. Thus, to link the primary and dual problem and enable the joint optimization, we introduce a cross-modal intermediate feature $\mathbf{f}_m$ to convey information of both the input of the primary problem $(V, E_o)$ and the dual problem $(V, M_o)$, as shown in Figure 2 (b). $\mathbf{f}_m$ is defined between the encoder and decoder stages of single-modal operations, i.e., $\Pi_v^{enc}, \Pi_e^{enc}, \Pi_v^{dec}, \Pi_e^{dec}$, to only focus on the multi-modal interaction.

**Relational cycle consistency.** A key observation for cycle consistency between $\mathcal{E}$ and $\mathcal{E}'$ is that the mapping between them is not necessarily bijective, as there could be multiple textual descriptions for the same object. Thus, naively adding point-wise consistency, i.e., $\mathbf{e}_o = \mathbf{e}'_o, \forall \mathbf{e}_o \in \mathcal{E}$ will collapse the feature space to a sub-optimal solution. Instead, we take inspiration from relational knowledge distillation [33], and introduce relational cycle consistency for $\mathcal{E}$ and $\mathcal{E}'$. The relational cycle consistency is to preserve the structure of the feature space rather than exact point-wise consistency, as illustrated in Figure 2 (c) and (d). Mathematically, the structure-preserving property is defined as isometric and conformal constraints to preserve pair-wise distance and angles for $\mathbf{e} \in \mathcal{E}$ and $\mathbf{e}' \in \mathcal{E}'$:

$$|\mathbf{e}_1 - \mathbf{e}_2| = |\mathbf{e}'_1 - \mathbf{e}'_2| \tag{2}$$
$$\angle(\mathbf{e}_1, \mathbf{e}_2, \mathbf{e}_3) = \angle(\mathbf{e}'_1, \mathbf{e}'_2, \mathbf{e}'_3), \tag{3}$$

where $|\cdot|$ and $\angle(\cdot)$ denote distance and angle metrics.

## 4  Method

In this section, we elaborate our $\mathrm{R}^2$-VOS framework with the relational consistency, which mainly consists of four parts: feature extraction, early grounding as a medium, video-text (V-T) projection for text reconstruction, and mask decoding for final segmentation, as shown in Figure 3. We first extract the video feature $\mathbf{f}$, word-level text feature $\mathbf{g}$, and sentence-level text embedding $\mathbf{e}$. On the one hand, to model the primary segmentation problem of maximizing $p(M_o|V, E_o)$, we enable the multimodal interaction in the early grounding module to generate the grounded feature $\mathbf{f_{early}}$. $\mathbf{f_{early}}$ coarsely locates the referred object $o$ and filters out irrelevant features, which serves as a medium linking the primary segmentation and dual text reconstruction problem. The final mask $M_o$ is obtained by dynamic convolution [4] on the decoded visual feature maps, with kernels learned from instance embedding $\{\mathbf{z}_t\}_{t=1}^{T}$. On the other hand, to model the dual text reconstruction problem of maximizing $p(E_o|V, M_o)$, we utilize the grounded video feature $\mathbf{f_{early}}$ as the alternative of $V$ and $M_o$, since $\mathbf{f_{early}}$ conveys contextual video clues of object $o$. In this way, we enable the parallel optimization of the primary and dual problem by relating them to $\mathbf{f_{early}}$. Specifically, we employ a V-T projection module to project $\mathbf{f_{early}}$ onto a reconstructed text embedding $\mathbf{e}'$. We add relational constraint between

139 $\mathbf{e}'$ and $\mathbf{e}$ to enforce the semantic alignment between the segmented mask and expression for positive
140 videos. In addition, we introduce a semantic consensus discrimination head $\mathcal{H}(\mathbf{e}, \mathbf{e}')$ to assess the
141 consistency between original and reconstructed text embeddings, discriminating the alignment of
142 multimodal semantics and identifying negative videos.

## 4.1 Single-modal Feature Extraction

**Visual encoder.** Following previous methods [1, 44, 43], we build the visual encoder with a visual
backbone and a deformable transformer encoder [52] on top of it. The extracted features from the
backbone are flattened, projected to a lower dimension, added with positional encoding [12], and
then fed into a deformable transformer encoder [52] similar to the previous method [44]. We denote
the multi-scale output of the transformer encoder as $\mathbf{F}$ and the low-resolution visual feature map from
the backbone as $\mathbf{f}$, where $\mathbf{f} \in \mathbb{R}^{T \times C_v \times \frac{H}{32} \times \frac{W}{32}}$, $C_v$ is the feature channel, $T$ is the video length and $H$
and $W$ are the original image size.

**Textual encoder.** We leverage a pre-trained linguistic model RoBERTa [27] to map the input textual
expression $E_o$ to a textual embedding space. The textual encoder extracts a sequence of word-level
text feature $\mathbf{g} \in \mathbb{R}^{C_e \times L}$ and a sentence-level text embedding $\mathbf{e} \in \mathbb{R}^{C_e \times 1}$, where $C_e$ and $L$ are the
dimension of linguistic embedding space and the expression length respectively.

## 4.2 Early Grounding

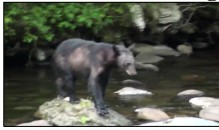 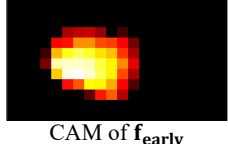

**a black bear standing on a rock in a stream**

Frame        CAM of $\mathbf{f_{early}}$

Figure 4: Visualization of channel activation map (CAM) of $\mathbf{f_{early}}$.

We propose an early grounding module to coarsely locate the referred object $o$ and filter out irrelevant features. Then the grounded feature $\mathbf{f_{early}}$ encoding information of $o$ can not only be utilized for the primary segmentation problem, but also for the dual expression reconstruction task, which serves as a proxy connecting the two problems. Figure 4 shows a visualization of $\mathbf{f_{early}}$. Specifically, we utilize the power of dynamic convolution [4] to discriminate visual features in the
early stage. As shown in the blue part of Figure 3, we first enable the multimodal interaction between
video and text features, then apply the dynamic convolution with kernels learned from text feature
to discriminate the object-level semantics. In particular, multi-head cross-attention (MCA) [41] is
leveraged to conduct the multimodal interaction:

$$\mathbf{h_f} = \text{LN}(\text{MCA}(\mathbf{f}, \mathbf{g}) + \mathbf{f}) \quad \mathbf{f}' = \text{LN}(\text{FFN}(\mathbf{h_f}) + \mathbf{h_f}) \tag{4}$$

$$\mathbf{h_g} = \text{LN}(\text{MCA}(\mathbf{g}, \mathbf{f}) + \mathbf{g}) \quad \mathbf{g}' = \text{LN}(\text{FFN}(\mathbf{h_g}) + \mathbf{h_g}), \tag{5}$$

where $\text{MCA}(\mathbf{X}, \mathbf{Y}) = \text{Attention}(\mathbf{W^Q X}, \mathbf{W^K Y}, \mathbf{W^V Y})$. $\mathbf{W}$ represents learnable weight. LN
and FFN denote layer normalization and feed-forward network respectively. The text feature $\mathbf{g}'$ is
further pooled to a fixed length, and followed by a fully-connected layer to form the dynamic kernels
$\Theta = \{\theta_i\}_{i=1}^{K}$. $K$ is the kernel number and $\theta_i \in \mathbb{R}^{C \times 1}$. The dynamic kernels are applied separately
to video feature $\mathbf{f}' \in \mathbb{R}^{C \times THW}$ to form the $\mathbf{f_{early}} \in \mathbb{R}^{C \times THW}$

$$\mathbf{f_{early}} = \text{BN}(\varphi(\theta_1^\text{T} \mathbf{f}' \oplus \cdots \oplus \theta_K^\text{T} \mathbf{f}') + \mathbf{f}'), \tag{6}$$

where $\oplus$ is the concatenation in channel dimension and $\varphi(\cdot)$ is a convolution to reduce the feature
dimension. BN denotes batch normalization.

## 4.3 Text Reconstruction

**V-T projection.** We leverage a transformer decoder $\mathcal{D}_E$ as textual decoder to transform the visual
representation of the referred object into the textual space. As shown in Figure 3, a learnable text
query $\mathbf{e}_0 \in \mathbb{R}^{C_e \times 1}$ is employed to query the $\mathbf{f_{early}}$. The output of the transformer decoder is the
reconstructed text embedding $\mathbf{e}' = \mathcal{D}_E(\mathbf{f_{early}}, \mathbf{e}_0) \in \mathbb{R}^{C_e \times 1}$.

## 4.4 Referring Segmentation

**Mask segmentation.** Similar to previous methods [44, 1, 11], we leverage deformable transformer
decoders with dynamic convolution to segment the object masks. As shown in Figure 3, we first fuse

the reconstructed text embedding $\mathbf{e}'$ to text embedding $\mathbf{e}$. The fused text embedding $\mathbf{e}$ is then repeated $N$ times to form the instance query [43] $\mathbf{z}_0 \in \mathbb{R}^{C_q \times N}$, where $C_q$ is the dimension of instance query and $N$ is the instance query number. We then use $T \times$ deformable transformer decoders $\mathcal{D}_V$ with shared weights to decode the instance embeddings $\mathbf{z}_t \in \mathbb{R}^{C_q \times N}$ for each frame, i.e., $\mathbf{z}_t = \mathcal{D}_V(\mathbf{F}_t, \mathbf{z}_0)$. $\mathbf{F}_t$ is the multiscale visual feature from visual encoder at time $t$. A dynamic kernel $\mathbf{w}_t$ is further learned from the instance embedding $\mathbf{z}_t$. The final feature map $\mathbf{f}_{\mathbf{out},t} \in \mathbb{R}^{C \times H \times W}$ is obtained by fusing low-level features from the feature pyramid network [23] in the visual decoder. The mask prediction $\mathbf{M}_t \in \mathbb{R}^{N \times H \times W}$ can be computed by $\mathbf{M}_t = \mathbf{w}_t^{\mathrm{T}} \mathbf{f}_{\mathbf{out},t}$.

**Auxiliary heads.** We build a set of auxiliary heads to obtain the final object masks across frames. In particular, a box head, a scoring head and a semantic consensus discrimination head are leveraged to predict the bounding boxes $\mathbf{B}_t \in \mathbb{R}^{N \times 4}$, confidence scores $\mathbf{S}_t \in \mathbb{R}^{N \times 1}$ and the alignment degree of multimodal semantics $A \in \mathbb{R}$. The box and scoring head are two fully-connected layers upon the instance embedding $\mathbf{e}_{\mathbf{t}}$. The semantic consensus discrimination head $\mathcal{H}(\mathbf{e}, \mathbf{e}')$ consists of two fully-connected layers upon the text embeddings $\mathbf{e} \oplus \mathbf{e}'$. Note that $A$ represents the semantic alignment in the entire video rather a single frame, since the expression is a video-level description.

## 4.5 Loss Function

The loss function of our method can be boiled down to three parts:

$$\mathcal{L} = \lambda_{text}\mathcal{L}_{text} + \lambda_{segm}\mathcal{L}_{segm} + \lambda_{align}\mathcal{L}_{align}, \tag{7}$$

where $\mathcal{L}_{text}$, $\mathcal{L}_{segm}$, and $\mathcal{L}_{align}$ are losses for text reconstruction, referring segmentation and semantic consensus discrimination respectively. A ground-truth semantic alignment $\hat{A} = \{0, 1\}$ is utilized to discriminate positive and negative pairs. The $\mathcal{L}_{align}$ is simply a cross-entropy loss between the predicted alignment $A$ and ground-truth $\hat{A}$. The other two terms are computed as follows:

**Loss for text reconstruction.** Given the text embedding $\mathbf{e}$ and reconstructed text embedding $\mathbf{e}'$, we employ a relational constraint to impose the cycle consistency between $\mathbf{e}$ and $\mathbf{e}'$. We calculate the loss by

$$\mathcal{L}_{text} = \mathbb{1}(\hat{A}) \cdot (\mathcal{L}_{dist} + \lambda_{angle}\mathcal{L}_{angle}), \tag{8}$$

where the indicator function $\mathbb{1}(\hat{A}) = 1$ if the alignment indicates the referred object exists in the video, otherwise 0, $\lambda_{angle}$ is a hyperparameter balancing the distance loss $\mathcal{L}_{dist}$ and angle loss $\mathcal{L}_{angle}$. We elaborate these two losses according to the relational cycle consistency Equation 2. Let $\mathcal{X}^n = \{(x_1, ..., x_n)|x_i \in \mathcal{X}\}$ denote a set of $n$-tuples, $\Phi^n = \{(\mathbf{x}, \mathbf{x}')|\mathbf{x} \in \mathcal{X}^n, \mathbf{x}' \in \mathcal{X}'^n\}$ denote a set of pairs consisting of two $n$-tuples of distinct elements from two different sets $\mathcal{X}$ and $\mathcal{X}'$. Specifically, the distance-based and angle-based relations relate text embeddings of 2-tuple and 3-tuple respectively, i.e., $\Phi^2 = \{(\mathbf{x}, \mathbf{x}')|\mathbf{x} = (\mathbf{e}_i, \mathbf{e}_j), \mathbf{x}' = (\mathbf{e}'_i, \mathbf{e}'_j), i \neq j\}$ and $\Phi^3 = \{(\mathbf{x}, \mathbf{x}')|\mathbf{x} = (\mathbf{e}_i, \mathbf{e}_j, \mathbf{e}_k), \mathbf{x}' = (\mathbf{e}'_i, \mathbf{e}'_j, \mathbf{e}'_k), i \neq j \neq k\}$. Then the losses are given by:

$$\mathcal{L}_{dist} = \sum_{(\mathbf{x}, \mathbf{x}') \in \Phi^2} l_\delta(\phi_D(\mathbf{x}), \phi_D(\mathbf{x}')), \quad \phi_D(\mathbf{x}) = \frac{1}{\mu(\mathbf{x})}\|\mathbf{e}_i - \mathbf{e}_j\|_2, \tag{9}$$

$$\mathcal{L}_{angle} = \sum_{(\mathbf{x}, \mathbf{x}') \in \Phi^3} l_\delta(\phi_\angle(\mathbf{x}), \phi_\angle(\mathbf{x}')), \quad \phi_\angle(\mathbf{x}) = \cos\angle(\mathbf{e}_i, \mathbf{e}_j, \mathbf{e}_k), \tag{10}$$

where $\mu(\mathbf{x}) = \sum_{\mathbf{x}=(x_1,x_2) \in \mathcal{X}^2} \frac{||x_1 - x_2||_2}{|\mathcal{X}^2|}$ is the average distance function, and the Huber loss $l_\delta(x, x') = \frac{1}{2}(x - x')^2$ if $|x - x'| \leq 1$, otherwise $|x - x'| - \frac{1}{2}$.

**Loss for referring segmentation.** Given a set of predictions $\mathbf{y} = \{\mathbf{y}_i\}_{i=1}^N$ and ground-truth $\hat{\mathbf{y}}$, where $\mathbf{y}_i = \{\mathbf{B}_{i,t}, \mathbf{S}_{i,t}, \mathbf{M}_{i,t}\}_{t=1}^T$ and $\hat{\mathbf{y}} = \{\hat{\mathbf{B}}_t, \hat{\mathbf{S}}_t, \hat{\mathbf{M}}_t\}_{t=1}^T$, we search for an assignment $\sigma \in \mathcal{P}_N$ with the highest similarity where $\mathcal{P}_N$ is a set of permutations of N elements ($\hat{\mathbf{y}}$ is padded with $\emptyset$). The similarity can be computed as

$$\mathcal{L}_{match}(\mathbf{y}_i, \hat{\mathbf{y}}) = \lambda_{box}\mathcal{L}_{box} + \lambda_{conf}\mathcal{L}_{conf} + \lambda_{mask}\mathcal{L}_{mask}, \tag{11}$$

where $\lambda_{box}$, $\lambda_{conf}$, and $\lambda_{mask}$ are weights to balance losses. Following previous works [6, 43], we leverage a combination of Dice [20] and BCE loss as $\mathcal{L}_{mask}$, focal loss [24] as $\mathcal{L}_{conf}$, and GIoU [38] and L1 loss as $\mathcal{L}_{box}$. The best assignment $\hat{\sigma}$ is solved by Hungarian algorithm [16]. Given the best assignment $\hat{\sigma}$, the segmentation loss between ground-truth and predictions is defined as $\mathcal{L}_{segm} = \mathbb{1}(\hat{A}) \cdot \mathcal{L}_{match}(\mathbf{y}, \hat{\mathbf{y}}_{\hat{\sigma}(i)})$.

### 4.6 Inference

During inference, we select the candidate with the highest confidence to predict the final masks:

$$\{\bar{\mathbf{M}}_t\}_{t=1}^T = \{\mathbb{1}(A) \cdot \mathbf{M}_{\bar{s},t}\}_{t=1}^T, \quad \bar{\mathbf{s}} = \underset{i}{\arg\max}\{\mathbf{S}_{i,1} + \cdots + \mathbf{S}_{i,T}\}_{i=1}^N, \tag{12}$$

where $\{\bar{\mathbf{M}}_t\}_{t=1}^T$ is the masks of referred object. $\mathbf{S}_{i,t}$ and $\mathbf{M}_{i,t}$ represent the $i$-th candidate in $\mathbf{S}_t$ and $\mathbf{M}_t$ respectively. $\bar{\mathbf{s}}$ is the slot with the highest confidence to be the target object. We use $\mathbb{1}(A)$ to filter out predictions in negative videos to mitigate false alarm. $\mathbb{1}(A) = 1$ if $A > 0.5$, else 0.

## 5 Experiment

### 5.1 Dataset and Metrics

**Dataset.** We conduct experiments on three datasets: Ref-Youtube-VOS, Ref-DAVIS and $\mathrm{R}^2$-Youtube-VOS. Ref-Youtube-VOS [39] is a large-scale benchmark that has 3,978 videos with about $15k$ language descriptions. There are 3,471 videos with 12,913 expressions in the training set and 507 videos with 2,096 expressions in the validation set. Ref-DAVIS-17 [13] contains 90 videos with 1,544 expressions, including 60 and 30 videos for training and validation respectively. $\mathrm{R}^2$-Youtube-VOS is our newly proposed evaluation dataset: it extends the Ref-Youtube-VOS validation set with each linguistic expression to query a positive video (the same one as Ref-Youtube-VOS) and a negative video. To make each video can be picked as a negative video, we randomly shuffle the original video set and constrain all negative text-video pair unrelated.

**Metrics.** We employ commonly-used region similarity $\mathcal{J}$ and contour accuracy $\mathcal{F}$ [36] for conventional Ref-Youtube-VOS and Ref-DAVIS-17 benchmarks. For the proposed $\mathrm{R}^2$-Youtube-VOS task, we additionally introduce a new metric $\mathcal{R} = 1 - \frac{\sum_{M \in \mathcal{M}_{neg}} |M|}{\sum_{M \in \mathcal{M}_{pos}} |M|}$ to evaluate the degree of object false alarm in negative videos, where $\mathcal{M}_{neg}$ and $\mathcal{M}_{pos}$ are the sets containing segmented masks in negative and positive videos respectively. $|M|$ denotes the foreground area of mask $M$. The total foreground area of positive videos $\sum_{M \in \mathcal{M}_{pos}} |M|$ serves as a normalization term. Ideally, a model should treat all the negative videos as backgrounds with $\mathcal{R} = 1$.

### 5.2 Implementation Details

Following previous methods [6, 44], our model is first pre-trained on Ref-COCO/+/g dataset [49, 31] and then finetuned on Ref-Youtube-VOS. The model is trained for 6 epochs with a learning rate multiplier of 0.1 at the 3rd and the 5th epoch. The initial learning rate is 1e-4 and a learning rate multiplier of 0.5 is applied to the backbone. We adopt a $batchsize$ of 8 and an AdamW [29] optimizer with weight decay $1 \times 10^{-4}$. Following convention [44, 1], the evaluation on Ref-DAVIS directly uses models trained on Ref-Youtube-VOS without re-training. All images are cropped to have the longest side of 640 pixels and the shortest side of 360 pixels during evaluation. The window size is set to 5 for all backbones. We create negative pairs by shuffling positive pairs in each batch. Our method is implemented with PyTorch [34].

### 5.3 Main Results

We compare our method with state-of-the-art R-VOS methods on Ref-Youtube-VOS and Ref-DAVIS-17 in Table 1, and $\mathrm{R}^2$-VOS task in Table 2.

**Comparison on Ref-Youtube-VOS.** In Table 1, we first compare our method on Ref-Youtube-VOS. For results of ResNet [9] backbone, our method achieves 57.3 $\mathcal{J}\&\mathcal{F}$ which outperforms the latest method ReferFormer [44] by 1.7 $\mathcal{J}\&\mathcal{F}$. In addition, our method runs at 30 FPS compared to 22 FPS of state-of-the-art ReferFormer (FPS is measured using single NVIDIA P40 with $batchsize = 1$). For results of Swin-Transformer [28, 28] backbones, our method achieves 60.2 $\mathcal{J}\&\mathcal{F}$ and 61.3 $\mathcal{J}\&\mathcal{F}$ with Swin-Tiny and Video-Swin-Tiny backbones respectively, which outperforms ReferFormer [44] and MTTR [1] by a clear margin. More analysis is available in the additional appendix A.1.

**Comparison on Ref-DAVIS-17.** Our method achieves 59.7 $\mathcal{J}\&\mathcal{F}$ on Ref-DAVIS-17 dataset, which outperforms ReferFormer by 1.2 $\mathcal{J}\&\mathcal{F}$.

| Method | Backbone | Ref-Youtube-VOS | | | Ref-DAVIS-17 | | |
|---|---|---|---|---|---|---|---|
| | | $\mathcal{J}\&\mathcal{F}$ | $\mathcal{J}$ | $\mathcal{F}$ | $\mathcal{J}\&\mathcal{F}$ | $\mathcal{J}$ | $\mathcal{F}$ |
| Spatial Visual Backbone | | | | | | | |
| CMSA [48] | ResNet-50 | 34.9 | 33.3 | 36.5 | 34.7 | 32.2 | 37.2 |
| CMSA + RNN [48] | ResNet-50 | 36.4 | 34.8 | 38.1 | 40.2 | 36.9 | 43.5 |
| URVOS [39] | ResNet-50 | 47.2 | 45.3 | 49.2 | 51.5 | 47.3 | 56.0 |
| PMINet [6] | ResNet-101 | 53.0 | 51.5 | 54.5 | - | - | - |
| CITD [22] | ResNet-101 | 56.4 | 54.8 | 58.1 | - | - | - |
| ReferFormer* [44] | ResNet-50 | 55.6 | 54.8 | 56.5 | 58.5 | 55.8 | 61.3 |
| **Ours** | ResNet-50 | **57.3** | **56.1** | **58.4** | **59.7** | **57.2** | **62.4** |
| ReferFormer* [44] | Swin-T | 58.7 | 57.6 | 59.9 | - | - | - |
| **Ours** | Swin-T | **60.2** | **58.9** | **61.5** | - | - | - |
| Spatio-temporal Visual Backbone | | | | | | | |
| MTTR* [1] | Video-Swin-T | 55.3 | 54.0 | 56.6 | - | - | - |
| ReferFormer* [44] | Video-Swin-T | 59.4 | 58.0 | 60.9 | - | - | - |
| **Ours** | Video-Swin-T | **61.3** | **59.6** | **63.1** | - | - | - |

Table 1: **Comparison to state-of-the-art R-VOS methods on Ref-Youtube-VOS and Ref-DAVIS-17 val set.** * indicates results imported from preprints.

| Method | Backbone | $\mathcal{J}\&\mathcal{F}$ & $\mathcal{R}$ | $\mathcal{J}$ | $\mathcal{F}$ | $\mathcal{R}$ |
|---|---|---|---|---|---|
| ReferFormer* [44] | ResNet-50 | 47.3 | 54.8 | 56.5 | 30.6 |
| **Ours** | ResNet-50 | **69.5** | **56.1** | **58.4** | **94.1** |
| MTTR* [1] | Video-Swin-T | 40.0 | 55.9 | 58.1 | 5.9 |
| ReferFormer* [44] | Video-Swin-T | 49.1 | 58.0 | 60.9 | 28.5 |
| **Ours** | Video-Swin-T | **72.8** | **59.6** | **63.1** | **95.7** |

Table 2: **Comparison to state-of-the-art R-VOS methods on $\mathrm{R}^2$-Youtube-VOS.**

**Comparison on $\mathrm{R}^2$-VOS.** As shown in Table 2, the state-of-the-art R-VOS methods, ReferFormer and MTTR suffer from a low $\mathcal{R}$ metric which measures the false-alarm problem when the semantic consensus of the input text-video pair does not hold. Compared to the severe false alarm of previous R-VOS methods, our model successfully mitigates the false alarm of the model, thanks to the proposed multimodal cycle consistency constraint and semantic consensus discrimination head.

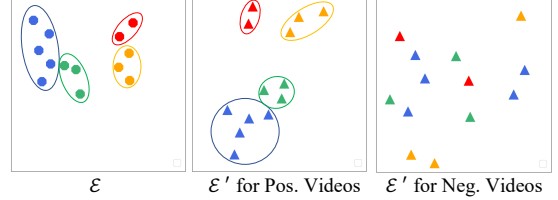

| $\mathcal{E}$ | $\mathcal{E}'$ for Pos. Videos | $\mathcal{E}'$ for Neg. Videos |

Figure 5: Visualization of text embedding spaces. Dots represent original text embeddings in $\mathcal{E}$, and triangles represent reconstructed ones in $\mathcal{E}'$ induced by positive and negative videos respectively. Elements in the same color belong to the same object. Note that an object can have multiple text descriptions. The structure of $\mathcal{E}'$ is well preserved from $\mathcal{E}$ for positive videos (ellipses bound embeddings of same objects), while it is not preserved for negative videos.

**Qualitative results.** We compare the qualitative results of our method against state-of-the-art methods in Figure 6 on $\mathrm{R}^2$-VOS. For **positive videos**: The result indicates that our method predicts accurate and temporally-consistent results, while Refer-Former [44] and MTTR [1] fail to locate the correct object. For **negative videos**: Both ReferFormer and MTTR suffer from a severe false-alarm problem when the referred object does not exist in the video. In contrast, with multi-modal cycle constraint and consensus discrimination, our method successfully filters out negative videos and mitigates the false alarm. To further explore how the consensus discrimination works, we visualize the text embedding and reconstructed text embedding spaces for both positive and negative videos. As shown in Figure 5, we notice that, for embeddings of positive videos, they preserve relative relations well, while for negative videos, the reconstructed embeddings have a random pattern in the space.

## 5.4 Ablation Study

**Module effectiveness.** To investigate the effectiveness of different components in our method, we conduct experiments with the ResNet-50 backbone on $\mathrm{R}^2$-Youtube-VOS dataset. We build a transformer-based baseline model and equip our proposed components step-by-step. As shown in Table 3, the baseline model achieves 52.4 $\mathcal{J}\&\mathcal{F}$. Then, we add our proposed components step-by-step to demonstrate the module effectiveness. After employing the early grounding module, the performance boosts to 55.5 $\mathcal{J}\&\mathcal{F}$ and the cycle-consistency constraint brings another 1.4 $\mathcal{J}\&\mathcal{F}$ gain. Since the reconstructed text embedding is generated with visual features injected, we consider it can

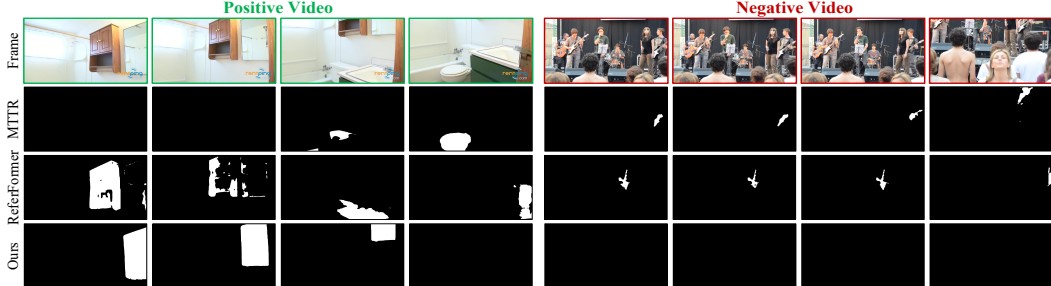

Expression: **the mirror in the bathroom is to the right of the wood cabinet**

Figure 6: Qualitative comparison to the state-of-the-art R-VOS method on the $R^2$-VOS task.

| Components | $\mathcal{J}\&\mathcal{F}$ | $\mathcal{J}$ | $\mathcal{F}$ | $\mathcal{R}$ |
|---|---|---|---|---|
| Baseline | 52.4 | 51.9 | 52.8 | 34.9 |
| +EG | $55.5_{+3.1}$ | 54.4 | 56.5 | $32.9_{-2.0}$ |
| +CC | $56.9_{+4.5}$ | 55.7 | 58.1 | $94.0_{+59.1}$ |
| +FT | $\mathbf{57.3}_{+4.9}$ | $\mathbf{56.1}$ | $\mathbf{58.4}$ | $\mathbf{94.1}_{+59.2}$ |

Table 3: **Impact of different components in our method.** EG: Early grounding, CC: Consistency constraint, FT: Fusing text embeddings.

| Constraint | $\mathcal{J}\&\mathcal{F}$ | $\mathcal{J}$ | $\mathcal{F}$ | $\mathcal{R}$ |
|---|---|---|---|---|
| None | 55.5 | 54.4 | 56.5 | 32.9 |
| PW | $54.4_{-1.1}$ | 53.3 | 55.5 | $88.7_{+55.8}$ |
| RA | $56.7_{+1.2}$ | 55.5 | 57.9 | $93.6_{+60.7}$ |
| RD | $56.4_{+0.9}$ | 55.2 | 57.6 | $90.4_{+57.5}$ |
| RD+RA | $\mathbf{56.9}_{+1.4}$ | $\mathbf{55.7}$ | $\mathbf{58.1}$ | $\mathbf{94.0}_{+61.1}$ |

Table 4: **Impact of the cycle consistency constraint.** PW: Point-wise. RA: Relational angle. RD: Relational distance.

| Query Number | $\mathcal{J}\&\mathcal{F}$ | $\mathcal{J}$ | $\mathcal{F}$ | $\mathcal{R}$ |
|---|---|---|---|---|
| 1 | 54.9 | 54.2 | 55.6 | $\mathbf{94.7}$ |
| 5 | $\mathbf{57.3}$ | 56.1 | $\mathbf{58.4}$ | 94.1 |
| 9 | 57.0 | $\mathbf{56.8}$ | 57.2 | 93.5 |

Table 5: **Impact of the query number.**

| Window Size | $\mathcal{J}\&\mathcal{F}$ | $\mathcal{J}$ | $\mathcal{F}$ | $\mathcal{R}$ |
|---|---|---|---|---|
| 1 | 53.5 | 53.0 | 54.0 | 89.2 |
| 3 | 56.8 | $\mathbf{56.5}$ | 57.1 | 92.1 |
| 5 | $\mathbf{57.3}$ | 56.1 | $\mathbf{58.4}$ | $\mathbf{94.1}$ |

Table 6: **Impact of the window size.**

encode some visual information, thus augmenting the original text embedding. By using the fused text embedding as instance query, we achieve our best performance of 57.3 $\mathcal{J}\&\mathcal{F}$.

**Consistency constraint.** We conduct experiments to ablate the influence of cycle-consistency constraints. As shown in Table 4, utilizing point-wise consistency constraint will lead to a performance drop compared to the setting without cycle constraint. We consider the point-wise constraint may force an injective mapping from the textual domain to the visual domain. However, the mapping can be a many-to-one function for R-VOS, i.e., each object corresponds to multiple textual descriptions. In addition, since the early grounding leverages the text feature to locate the referred object, if we use the direct point-wise constraint to form reconstructed text embedding, it will encourage the network to memorize the text feature in the $\mathbf{f_{early}}$ and result in a collapse for text reconstruction. Table 4 shows that sole relational angle constraint can bring 1.2 $\mathcal{J}\&\mathcal{F}$ gain, and it can be slightly improved with 1.4 $\mathcal{J}\&\mathcal{F}$ gain by jointly using relational angle and distance constraint.

**Instance query number.** Although only one referral is involved for each frame in R-VOS task, to help the network optimization, we employ more than one instance query to each video. Table 5 indicates that a query number of 5 brings the best result.

**Frame number.** Since R-VOS gives a text that describes an object over a period of time, temporal information is vital to segment accurate and temporally-consistent results. We ablate on the best window size of input videos during training. As shown in Table 6, we notice that the performance improves as the window size increases and a window size of 5 brings the best result of 57.3 $\mathcal{J}\&\mathcal{F}$.

# 6 Conclusion

In this paper, we investigate the semantic misalignment problem in R-VOS task. A pipeline jointly models the referring segmentation and text reconstruction problem, equipped with a relational cycle consistency constraint, is introduced to discriminate and enhance the semantic consensus between visual and textual modalities. To evaluate the model robustness, we extend the R-VOS task to accept unpaired inputs and collect a corresponding $R^2$-Youtube-VOS dataset. We observe a severe false-alarm problem suffered from previous methods on $R^2$-Youtube-VOS while ours accurately discriminates unpaired inputs and segments high-quality masks for paired inputs. Our method achieves state-of-the-art performance on Ref-DAVIS17, Ref-Youtube-VOS, and $R^2$-VOS dataset. We believe that, with unpaired inputs, $R^2$-VOS is a more general setting of referring video segmentation, which can shed light on a new direction to investigate the robustness of referring segmentation.

# A  Additional Appendix

## A.1  More Quantitative Result Analysis

Under the same ResNet-50 backbone, our method achieves 57.3 $\mathcal{J}\&\mathcal{F}$, 94.1 $\mathcal{R}$ and 30 FPS compared to the 55.6 $\mathcal{J}\&\mathcal{F}$, 30.6 $\mathcal{R}$ and 22 FPS of ReferFormer. We will then point-to-point analyze reasons of improvements on $\mathcal{J}\&\mathcal{F}$ (for positive video), $\mathcal{R}$ (for negative videos) and FPS (for inference speed).

- $\mathcal{J}\&\mathcal{F}$: (1) We introduce the early-grounding module which employs both pixel-wise and channel-wise attention to enable multimodal interaction. Different from the CM-FPN used in ReferFormer that solely fuses features from text to video in pixel-level, our early-grounding module first enables pixel-level bi-directional fusion and then generates dynamic kernels using the fused text feature $\mathbf{g}'$ to modulate the video feature $\mathbf{f}'$. The dynamic convolution (channel-wise attention) is commonly used to decode dense masks from visual features and is suitable to suppress irrelevant features. By equipping text-guided dynamic convolution in early-stage, the pixel decoder can be more focused on the target object (as shown in Figure 4). (2) Our method leverages relational cycle consistency to constraint the intermediate feature $\mathbf{f}_{\text{early}}$ to contain correct object-level information to recover some properties of original text embedding. By applying this constraint, our method can better avoid interference and easier locate the correct object. (3) Our instance query is composed of both original sentence embedding and the reconstructed one. Different from ReferFormer that only utilizes original sentence embedding as queries, the reconstructed embedding can encode visual information to facilitate the instance query decode the objects from visual features.

- $\mathcal{R}$: The newly introduced metric $\mathcal{R}$ aims to measure the robustness of the model against unpaired inputs. Text-video pairs with (object-level) semantic consensus can be assumed as in-distribution for RVOS problem where semantic consensus can be kind of easily modeled. In contrast, unpaired text-video is much more difficult to tackle because there can be unlimited out-of-distribution (OOD) scenarios for the text-video pairs. In our method, instead of directly detect the OOD of input pairs, we convert the problem to find semantic alignment between the input text embedding and reconstructed embedding and constraint the property of reconstructed space by introducing the cycle consistency. In this way, the comparison is conducted in the constraint original and reconstructed text spaces. For ReferFormer, it models the alignment of text to video by querying the visual features by text in the transformer decoder. In this way, the comparison is conducted in unconstrained text and video spaces thus results in a inferior performance.

- FPS: The speed improvement of our method mainly comes from our efficient multimodal fusion. Compared to the multi-scale CM-FPN, our early-grounding module is only conduct at the high-level. In addition, our bi-direction multimodal fusion (Equ 4 & 5) only leverages cross-attention to avoid computational expensive video-to-video operations.

## A.2  Limitations

An important challenge for video segmentation is that target object disappearance due to occlusion, which can results in false positives on a per-frame level. In our method, we predict the video-level semantic alignment to handle the false positive in video-level resulted from unpaired text-video pairs. However, since only video-level object expression is available in RVOS task, our method can not address the frame-level false positives resulted from occlusion.

## A.3  Additional Experiment on Negative Videos without Positive Text

| Negative Video Source | $\mathcal{R}$ | |
|---|---|---|
| | ReferFormer | Ours |
| Ref-Youtube-VOS | 30.6 | 94.1 |
| Ref-Youtube-VOS & Ref-DAVIS | 33.1 | 92.2 |

Table A: Impact of different negative video sources.

As shown in Table A, we test the robustness of our model on two settings. We generate negative videos from Ref-Youtube-VOS and a combination of Ref-Youtube-VOS and Ref-DAVIS dataset.

In both settings, all videos in the validation set are leveraged. The results indicates that source of negative videos has minor impact on the robustness of our model.

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
