# OpenReview forum: "R^2-VOS: Robust Referring Video Object Segmentation via Relational Cycle Consistency"
_NeurIPS.cc/2022/Conference — NeurIPS 2022 Submitted_

### Official Review · Reviewer_Gh1L · 2022-07-09

**Rating:** 6
**Confidence:** 4
**Soundness:** 3 good
**Presentation:** 3 good
**Contribution:** 3 good

**Summary:**

This paper introduces the Robust R-VOS task that accepts unpaired video and text as inputs. In addition, a pipeline is proposed to optimize the primary referring and dual expression reconstruction task. A relational cycle consistency constraint is proposed to enhance the semantic alignment between different modalities. The proposed approach achieves state-of-the-art results on three datasets.

**Questions:**

1 In formula 8, l(A ̂) is used to measure semantic consensus between Reconstructed text feature and text feature, it should be explained clearly.
2 It is not clearly stated in the article how to evaluate this model on the Robust R-VOS dataset. Is the model trained on the Ref-Youtube-VOS dataset and directly evaluated on the Robust R-VOS dataset?
3 The ablation experiments in this paper are performed on the Ref-Youtube-VOS dataset, which does not need to consider this problem that unpaired video and text as inputs, so different modules need be used separately to verify their effectiveness.
4 The authors should clearly introduce the source of the negative sample videos in the dataset.  If these videos are also from the Ref-Youtube-VOS dataset, each video in the entire dataset will correspond to a positive text message. In order to better verify the robustness of the model, video that do not correspond to text information in the entire dataset should be added.


**Limitations:**

Yes

**Strengths And Weaknesses:**

Strengths:
1)This paper proposed the Robust R-Youtube-VOS dataset, is collected to measure the robustness of R-VOS models.
2)The relational cycle consistency is applied to preserve the structure of the feature space rather than point-wise consistency. Through the experience, the cycle consistency achieves better results than point-wise consistency.
3)The proposed approach achieves state-of-the-art on three datasets, and some ablation studies are conducted to validate the effectiveness of each component.
4)The paper is generally well written.

Weaknesses:
1) My main concern is that the main novelty. In this paper, for negative videos
M_o is empty. For some R-VOS methods, they are able to align the semantic information of different modalities, and they also overcome the problem that unpaired video and text as inputs so that Robust R-VOS does not need to be studied separately.
2) This paper proposed a new task and proposed a new dataset. However, the paper makes the ablation study based on the Ref-DAVIS17, and Ref-Youtube-VOS.
3) This paper leverages text reconstruction to reconstruct the text feature to measure semantic consensus. It is commonly used for Video-Language tasks.

---

> ### Author Response · Authors · 2022-08-01
> **Response to Reviewer Gh1L - Part 2 for Questions**
>
> Here are answers to the raised questions.
>
> ---
> **4. Detailed explanation for $1(\hat{A})$ in Equ 8.**
>
> In Equation 8, $\hat{A}$ denotes the ground-truth semantic alignment degree, i.e., the probability indicating whether the text expression corresponds to the visual object in the video, which equals to 1 for paired input and 0 for unpaired ones. $1(\hat{A})$ aims to filter out the text reconstruction loss $\mathcal{L}_{text}$ for unpaired inputs, since text-video-text cycle is meaningless.
> In addition, we note that $A \in [0,1]$ is the predicted alignment degree. The indicator function 1(A) equals to 1 when $A>0.5$, else 0.
>
>
> ---
> **5. Is the model trained on the Ref-Youtube-VOS dataset and directly evaluated on the $\mathrm{R}^2$-VOS dataset?**
>
> Yes. We train all the models based on the original Ref-Youtube-VOS dataset. $\mathrm{R}^2$-VOS dataset is an evaluation set.
>
> * Although using the same Ref-Youtube-VOS dataset in training, we augment it with both positive and negative text-video pairs, which is different from previous R-VOS methods. We construct negative pairs in each batch. After each iteration of positive pairs, we shuffle the video set in the batch to create negative pairs and feed them into the network again. The ground-truth semantic alignment $\hat{A}$ is set to 1 for positive pairs and 0 for negative pairs. We will elaborate detailed training in the revision.
>
> * The evaulation dataset $\mathrm{R}^2$-VOS can be assumed as an augmented Ref-Youtube-VOS val set containing two parts: Ref-Youtube-VOS val set (for positive videos, evaluated by $\mathcal{J}$\&$\mathcal{F}$) and shuffled Ref-Youtube-VOS val set (for negative videos, evaluated by $\mathcal{R}$).
>
> ---
> **6. The source of the negative sample videos in the dataset. Video that do not correspond to text information in the entire dataset should be added.**
>
> For negative pairs in $\mathrm{R}^2$-VOS dataset, the videos and texts are still selected from the Ref-Youtube-VOS validation set, while the order of videos is shuffled to make sure that each text queries to a different video.
>
> To evaluate on videos that are not in Ref-Youtube-VOS, we collect **new negative videos** by adding videos in Ref-DAVIS val set into the original Ref-Youtube-VOS val set. We report the $\mathcal{R}$ metric in the table below. This evaluation indicates that the source of negative videos does not have high impact on the robustness of our model.
>
> |Negative Video Source|ReferFormer ($\mathcal{R}$)|Ours ($\mathcal{R}$)|
> | ---- | ---- | ---- |
> |Ref-Youtube-VOS|30.6|94.1|
> |Ref-Youtube-VOS + Ref-DAVIS|33.1|92.2|

---

> ### Author Response · Authors · 2022-08-01
> **Response to Reviewer Gh1L - Part 1 for Weakness**
>
> We thank the reviewer for the valuable comments. Our answers are split into two parts: Part 1 for weakness and Part 2 for Questions.
>
> ---
> **1. Some R-VOS methods overcome the problem that unpaired video and text as inputs so that $\mathrm{R}^2$-VOS does not need to be studied separately.**
>
> We argue that studying the $\mathrm{R}^2$-VOS problem is necessary.
> To the best of our knowledge, there are no existing R-VOS methods that can overcome the false positives caused by unpaired inputs. Please be aware of the robustness $\mathcal{R}$ evaluation in Table 2 and qualitative results in Figure 6 (negative pairs), the previous state-of-the-art methods, ReferFormer and MTTR, can not handle such cases well.
> Although existing R-VOS methods align multi-modal semantic information by learning with positive text-video pairs, they didn't consider the negative pairs. Negative pairs are demonstrated helpful to learn correspondences between two feature spaces [1]. Our model exploits the negative pairs in training and achieve better robustness.
>
> In summary, we propose the new $\mathrm{R}^2$-VOS problem to generalize and refine the original R-VOS problem from the data, methodology and evaluation aspects: augment both the training and validation datasets with negative pairs; enhance or verify multi-modal consistency according to (object-level) semantic consensus with cycle consistency; introduce a new metric of robustness $\mathcal{R}$ against unpaired inputs for practical scenarios.
>
> [1]  Learning transferable visual models from natural language supervision, ICML 2021
>
> ---
> **2. This paper proposed a new task and proposed a new dataset while makes the ablation study based on the Ref-Youtube-VOS.**
>
> Ablation study on the $\mathrm{R}^2$-VOS dataset is a very constructive suggestion, and we update new $\mathcal{R}$ metrics on $\mathrm{R}^2$-VOS in the following tables (also marked in blue in the revised submission). As the positive video evaluation of $\mathrm{R}^2$-VOS is the same as Ref-Youtube-VOS, the numbers of $\mathcal{J}$\&$\mathcal{F}$ do not change.
>
> Table3 Impact of different components (note that components are added step by step):
>
> | Components | $\mathcal{J}$\&$\mathcal{F}$ | $\mathcal{J}$ | $\mathcal{F}$ | $\mathcal{R}$ |
> | ---- | ---- | ---- | ---- | ---- |
> | Baseline | 52.4 | 51.9 | 52.8 | 34.9 |
> | +EG | 55.5$_{\rm{\bf{+3.1}}}$ | 54.4 | 56.5 | 32.9$_{\rm{\bf{-2.0}}}$ |
> | +CC | 56.9$_{\rm{\bf{+4.5}}}$ | 55.7 | 58.1 | 94.0$_{\rm{\bf{+59.1}}}$ |
> | +FT | 57.3$_{\rm{\bf{+4.9}}}$ | 56.1 | 58.4 | 94.1$_{\rm{\bf{+59.2}}}$ |
>
>
> Table 4 Impact of cycle consistency constraint:
>
> | Constraint | $\mathcal{J}$\&$\mathcal{F}$ | $\mathcal{J}$ | $\mathcal{F}$ | $\mathcal{R}$ |
> | ---- | ---- | ---- | ---- | ---- |
> | None | 55.5 | 54.4 | 56.5 | 32.9 |
> | PW | 54.4$_{\rm{\bf{-1.1}}}$ | 53.3 | 55.5 | 88.7$_{\rm{\bf{+55.8}}}$ |
> | RA | 56.7$_{\rm{\bf{+1.2}}}$ | 55.5 | 57.9 | 93.6$_{\rm{\bf{+60.7}}}$ |
> | RD | 56.4$_{\rm{\bf{+0.9}}}$ | 55.2 | 57.6 | 90.4$_{\rm{\bf{+57.5}}}$ |
> | RD+RA | 56.9$_{\rm{\bf{+1.4}}}$ | 55.7 | 58.1 | 94.0$_{\rm{\bf{+61.1}}}$ |
>
> Table 5 Impact of the query number:
>
> | Query Number | $\mathcal{J}$\&$\mathcal{F}$ | $\mathcal{J}$ | $\mathcal{F}$ | $\mathcal{R}$ |
> | :----: | ---- | ---- | ---- | ---- |
> | 1 | 54.9 | 54.2 | 55.6 | 94.7 |
> | 5 | 57.3 | 56.1 | 58.4 | 94.1 |
> | 9 | 57.0 | 56.8 | 57.2 | 93.5 |
>
> Table 6 Impact of the window size:
> | Window Size | $\mathcal{J}$\&$\mathcal{F}$ | $\mathcal{J}$ | $\mathcal{F}$ | $\mathcal{R}$ |
> | :----: | ---- | ---- | ---- | ---- |
> | 1 | 53.5 | 53.0 | 54.0 | 89.2 |
> | 3 | 56.8 | 56.5 | 57.1 | 92.1 |
> | 5 | 57.3 | 56.1 | 58.4 | 94.1 |
>
> ---
> **3. Text reconstruction to reconstruct the text feature to measure semantic consensus is commonly used for Video-Language tasks.**
>
> We agree that utilizing text reconstruction in Video-Language tasks is common, but our work is very different from the previously used text-vision-text cycle consistency, also stated in Line 97-124.
> * (1) We use **relational** cycle consistency instead of the previous **point-wise** counterpart, making the cycle constraint feasible between two feature spaces that do not have strict bijective mapping (Figure 2 (d)). In particular, the mapping from visual objects to textual expressions is not necessarily bijective, as there could be multiple textual descriptions for the same object (about 5 for Ref-Youtube-VOS). Thus, naively adding point-wise consistency may make the feature space collapse (Line 116-119). Our ablation study in **Table 4** demonstrates the effectiveness of our relational cycle consistency. The point-wise cycle consistency even decreases the accuracy.
> * (2) We apply the cycle consistency on the text embedding space instead of the text expression space, which avoids the dataset bias of the pretrained linguistic model from other datasets. Also, we enable the joint optimization of the primary and dual problem efficiently without decoding text embeddings into expressions, as illustrated in Figure 2 (b).

---

> > ### Comment · Reviewer_Gh1L · 2022-08-08
> > **post-rebuttal response**
> >
> > Thanks for reporting more experimental results. I still have several issues after reading the rebuttal.
> > 1. For Table3 Impact of different components, the ablation study is too simple by removing each component independently. The underlying contribution between CC and FF  is not clear. More in-depth analysis about these two components should be discussed, such as adding them step-by-step.
> > 2. As Reviewer Fhve pointed out, the difference between this work and RefFormer should be further clarified.

---

> > > ### Author Response · Authors · 2022-08-08
> > > **Response to Reviewer Gh1L**
> > >
> > > Thanks for your further suggestions! Our answers to the questions are as follows.
> > >
> > > ---
> > > **1. The ablation study is too simple by removing each component independently.**
> > >
> > > Sorry for the misleading notions in Table 3. Our ablation study is conducted by adding each component step by step (Line 302-303 in the revised paper). The  **gains** reported in Table 3 are based on the baseline setting. The **relative gain** for FT is 0.4 $\mathcal{J}$\&$\mathcal{F}$ and 0.1 $\mathcal{R}$. The contribution of FT is not significant. We have revised the notion of Table 3 to clarify the underlying relation between each experiment.
> > >
> > > Table3: Impact of different components in our method:
> > >
> > > | Components | $\mathcal{J}$\&$\mathcal{F}$ | $\mathcal{J}$ | $\mathcal{F}$ | $\mathcal{R}$ |
> > > | ---- | ---- | ---- | ---- | ---- |
> > > | Baseline | 52.4 | 51.9 | 52.8 | 34.9 |
> > > | +EG | 55.5$_{\rm{\bf{+3.1}}}$ | 54.4 | 56.5 | 32.9$_{\rm{\bf{-2.0}}}$ |
> > > | +EG+CC | 56.9$_{\rm{\bf{+4.5}}}$ | 55.7 | 58.1 | 94.0$_{\rm{\bf{+59.1}}}$ |
> > > | +EG+CC+FT | 57.3$_{\rm{\bf{+4.9}}}$ | 56.1 | 58.4 | 94.1$_{\rm{\bf{+59.2}}}$ |
> > >
> > > ---
> > > **2. The difference between this work and ReferFormer.**
> > >
> > > We have clarified the differences between our paper and ReferFormer in Response to Reviewer fwGH, and also added the Appendix.A in the revised paper. Here we list the discussion:
> > >
> > > We summarize differences between our work and ReferFormer, and will add the discussion in the revision.
> > >
> > > - Different from all the existing R-VOS methods, including ReferFormer, using all positive text-video pairs for training, we use both positive and negative pairs, which help the learning of differentiating semantic consensus between different pairs.
> > >
> > > - We leverage the relational text-video-text cycle consistency to better correspond the text embedding space to the video embedding space. Positive pairs are constrained with the cycle consistency for better embedding learning, while negative pairs unconstrained with the cycle constraint could be identified.
> > >
> > > - We utilize the early-grounding module, which modulates the video feature with the video-aware text embedding. Thus, irrelevant video features are suppressed in an early stage, while ReferFormer only uses dynamic convolution in the final mask decoding stage, easier to involve irrelevant objects, as shown in the results of positive pairs Figure 6.
> > >
> > > - Our instance query is composed of both the original sentence embedding and the reconstructed one. Different from ReferFormer that only utilizes original sentence embedding as queries, the reconstructed embedding can encode visual information to facilitate the instance query to decode the objects from visual features.
> > > We have already shown more qualitative results in the video demo of the supplemental material, and will add some samples in the main paper.
> > >
> > > - Our method achieves superior performance than Referformer. Under the same ResNet-50 backbone, our method achieves 57.3 $\mathcal{J}$\&$\mathcal{F}$, 94.1 $\mathcal{R}$ and 30 FPS compared to the 55.6 $\mathcal{J}$\&$\mathcal{F}$, 30.6 $\mathcal{R}$ and 22 FPS of ReferFormer (more analysis to improvements on each metric is available in Appendix.A).

---

### Official Review · Reviewer_Fhve · 2022-07-14

**Rating:** 4
**Confidence:** 3
**Soundness:** 3 good
**Presentation:** 3 good
**Contribution:** 2 fair

**Summary:**

The authors introduce a "new" variant of referring video object segmentation. This variant considers the possiblity that the given referring expression does not correspond to any object in the given video. A method based on relational cycle consistency is introduced as well. The proposed method has an additional head that predicts whether there is an object that corresponds to the given expression or not. Experiments are mainly conducted on two standard benchmarks to validate the effectiveness of the proposed method.

**Questions:**

What is the range of A? When does the indicator function 1(A) equal to 1?

**Limitations:**

N.A.

**Strengths And Weaknesses:**

Strengths:
(1) The paper is well written is easy to understand. Notations and symbols are well defined.
(2) The authors propose a method that can handle expression that does not correspond to any objects in the given video via a simple head, which is likely a simple fully connected layer.
(3) The proposed method is evaluated on two standard benchmarks and achieves good performance on both datasets.

Weakness:
(1) While existing methods do not explicitly consider the possiblilty of being given an expression that does not correspond to any objects in the video, it is likely that this can be easily handled by adding a background class to ReferFormer [1] and other methods.
(2) Researchers have already used cycle consistency (text reconstruction) to help referring expersion segmentation in images. To me, this work is mainly an extension of existing ideas to video.

[1] Language as Queries for Referring Video Object Segmentation, CVPR'22

---

> ### Author Response · Authors · 2022-08-01
> **Response to Reviewer  Fhve**
>
>
> We thank the reviewer for the time and effort to review our paper. Our answers to the questions are as follows.
>
> ---
> **1. False positives can be easily handled by adding a background class to ReferFormer and other methods.**
>
> We would like to argue that simply adding a background class in existing methods, like ReferFormer, cannot handle the cases of an expression that does not correspond to any objects in the video well.
> In fact, ReferFormer has already involved a background (empty) class to indicate the object existence at the frame level (in the Section 3.4 of the ReferFormer paper, represented as $p_i^t$). Similarly, MTTR also has a background class. However, the robustness evaluation in Table 2 indicates that the background class with a simple classification head cannot effectively discriminate semantic consensus between input pairs.
>
>
> Here are the reasons.
> Existing R-VOS methods only use positive text-video pairs in training, while none of negative pairs are sampled. Modeling negative correspondences between textual expressions and visual object is an out-of-distribution (OOD) problem as any unrelated expressions to the video can be regarded as a negative sample. Existing methods only consider positive pairs, and it is known that a naive classification head cannot solve the OOD problem well. Thus, modeled correspondences between unpaired textual expressions and videos by these methods could be uncertain, usually resulting in false positives, as Figure 6 shown.
> Unlike some previous methods (ReferFormer, MTTR) directly applying a classification head on the output of text-to-video queries, our model exploit the consistency in the text-to-video-to-text cycle and enable the discrimination by assessing the cycle consistency, which smartly circumvent difficulties in the classification for OOD problem. We also add negative pairs in training for learning better discrimination.
>
>
> ---
> **2. Researchers have already used cycle consistency (text reconstruction) to help referring expression segmentation in images. It is an extension to videos.**
>
> Our method is exactly not an extension of the conventional cycle consistency used in previous referring image segmentation methods [1,2].
> As stated in Line 97-124, our method is very different from the previously used cycle consistency:
> * (1) We use **relational** cycle consistency instead of the previous **point-wise** counterpart, which makes the cycle constraint feasible between two feature spaces that do not have strict bijective mapping, as illustrated in Figure 2 (d). In particular, the mapping from visual objects to textual expressions is not necessarily bijective, as there could be multiple textual descriptions for the same object (about 5 for Ref-Youtube-VOS). Thus, naively adding point-wise consistency may make the feature space collapse (Line 116-119). Our ablation study in **Table 4** demonstrates the effectiveness of our relational cycle consistency. The point-wise cycle consistency even decreases the accuracy.
> * (2) We apply the cycle consistency in the text embedding space instead of the original text expression space [1,2], which avoids the dataset bias of the pretrained linguistic model from other datasets. Also, we enable the joint optimization of the primary and dual problem efficiently without decoding text embeddings into expressions, as illustrated in Figure 2 (b).
>
>
> [1] Query reconstruction network for referring expression image segmentation, TMM 2020
>
> [2] Referring Expression Object Segmentation with Caption-Aware Consistency, arXiv 2019
>
> ---
> **3. What is the range of A? When does the indicator function 1(A) equal to 1?**
>
> The predicted alignment degree $A \in [0,1]$ is the probability indicating whether the textual expression corresponds to the visual object in the video. The indicator function $1(A)$ equals to 1 when $A>0.5$, else 0.

---

> > ### Comment · Reviewer_Fhve · 2022-08-09
> > **Post-rebuttal response**
> >
> > Thanks for the detailed response.
> >
> > **1. Handle unmatched experssions by adding a background class to ReferFormer and other methods.**
> >
> > As you pointed out, ReferFormer and other methods, _e.g._, MTTR, only train with positive samples, _i.e._, matched experssions. I agree with you that this is the reason why they do not work well on your dataset even if they have a background class. However, to me, this shows they have great potential to work well if trained with negative samples, _i.e._, unmatched experssions in the same way that your method is trained. In other words, your method is trained with samples that are not used to train ReferFormer and MTTR, thus making the comparison somewhat unfair.
> >
> > If you do not use any negative samples during training, they are OOD samples. However, if the negative samples are used during training, they are not OOD samples anymore. The negative samples could be handled by the background class of ReferFormer and MTTR, if you use them during training. This further validates my point that the comparison is not fair.
> >
> > **2. Novelty of the relational cycle consistency.**
> >
> > I agree that the proposed relational cycle consistency is not the same as existing ones. However, to me, a variant of the cycle consistency is not novel enough for me to change my rating.

---

> > > ### Author Response · Authors · 2022-08-09
> > > **Response to Reviewer Fhve**
> > >
> > > Thanks for your comments.
> > >
> > > ---
> > > **1. Handle unmatched expressions by adding a background class to ReferFormer and other methods.**
> > >
> > > We provide a further discussion to address that adding a naïve classification model that treats negative samples as an additional class is limited, while our method exploiting the nature of the problem with cycle consistency could handle the unmatched expressions.
> > >
> > > The principal difference is that between  **implicit** and **explicit** classes.  In the absence of negative samples, a "none of the above" (background class) is effectively an **implicit** class.  Being implicit, there are no training data provided for it, so you are correct, we end up handling it as a problem of trying to identify OOD through thresholding criteria.  There is a key feature here.  In OOD determination, there is no **discriminative** component of the model assigned to the class -- the rejection is effectively performed based entirely on low likelihood as computed from the distributions of the known classes, and as a consequence heuristics must be imposed.
> > >
> > > When we convert "none of the above" to an **explicit** class, as we have, it converts this to a discriminative modeling problem. The challenge is that, given the vast scope of the "none of the above" class, it is generally infeasible to obtain sufficient training data to model all possibilities.  This is a known problem.
> > >
> > > This has also been noticed in the ReferFormer and MTTR, where, when we introduce the none-of-the-above as an explicit class through a classification head, it provides no benefit -- the ReferFormer is unable to model it well.
> > >
> > > Our cyclic consistency approach provides us a way to capture this class  using just a limited number of training samples from this now-explicit class, and we are able to do this because of the specific nature of the R-VOS problem.  This, in fact, is a **novelty** of our approach -- we are exploiting the nature of the problem to be able to model this very diverse class effectively using a limited number of training samples.  This also clearly shows up in the performance numbers.
> > >
> > > As such, our approach is not a trivial increment.  We have proposed a solution that exploits the structure of the problem to achieve what a naive inclusion of negative samples in techniques such as ReferFormer would not.
> > >
> > > The reviewer is, unfortunately, dismissing our feature -- the introduction of an **explicit** class and a mechanism that allows us to handle its diversity as a solution that could hypothetically also have been achieved by other models, had they too introduced it as an explicit class.  While this may or may not be true (given that there is no evidence for or against it),  it should ideally not be held as a criterion for evaluation.  Otherwise, similar hypothetical criteria could be applied against most innovations.
> > >
> > > We thank the reviewer in advance for consideration of this fact.
> > >
> > > ---
> > > **2. Clarification of the novelty**
> > >
> > > We agree that cycle consistency is commonly used and has many variants. However, the novelty of our work is not only proposing the relational cycle consistency. We summarize our key novelty as follows.
> > >
> > > - Given the fact that referring expression is not always paired with the video in practice, we introduce the $\mathrm{R}^2$-VOS problem, which extends the R-VOS task to accept unpaired video and text as inputs, and propose a proper metric $\mathcal{R}$ to measure the false-positive objects.
> > > - We propose a pipeline that jointly optimizes the primary referring segmentation and dual expression reconstruction task. A early grounding module is introduced which serves as an intermediate proxy for both problems.
> > > - We introduce relational cycle consistency constraint to enhance the semantic alignment between visual and textual modalities. In particular, the constraint is applied in the embedding space to avoid decoding the textual embedding to expression (always biased by pretrained language model).
> > > - The proposed components not only help suppress false positives but also improve the segmentation ability. Our method achieves state-of-the-art performance on two common R-VOS dataset (Ref-Youtube-VOS, Ref-DAVIS), and our proposed  $\mathrm{R}^2$-Youtube-VOS

---

### Official Review · Reviewer_fwGH · 2022-07-21

**Rating:** 7
**Confidence:** 3
**Soundness:** 4 excellent
**Presentation:** 3 good
**Contribution:** 3 good

**Summary:**

This paper presents a new method for referring video object segmentation (R-VOS). It points out an assumption that current R-VOS datasets make: that the object referred to in the text is present in the video. This assumption is not true in real-world text-video query applications. Current models rely on the assumption which results in false positives when the referred object is not in the video. The paper addresses this in two ways: by introducing a dataset $R^{2}$-YouTube-VOS and evaluation metric that measures the false positives made by a model, and by creating a method that enforces the semantic matching of visual and text features. The authors demonstrate that their method outperforms current methods on Ref-YouTube-VOS and Ref-DAVIS and has fewer false positives than previous methods on the proposed $R^{2}$-YouTube-VOS.

**Questions:**

Questions:
- In the sentence in lines 242-243 can you clarify what "constrain all negative text-video pair unrelated" means?

Suggestions:
- There are many works on vision-language representation learning and it would be useful to the reader if the authors could include more of this background in Section 2.
- Line 58 a better word instead of "medium" would be "proxy"
- It would be more clear to call $f_{m}$ an intermediate feature rather than a medium in line 112.
- Typo in line 109 "textural" -> "textual"
- Figure 2c caption is a run-on sentence and would sound better split into two.
- When discussing the window size in line 259 mention that Swin or VideoSwin is used as a backbone.
- Refer to Figure 4 somewhere in the text.
- Line 320 "temporal-consistent" -> "temporally-consistent"
- The discussion of quantitative results is currently limited. It would benefit from more details on how the proposed method differs from the ReferFormer architecture and why the ReferFormer is not able to learn the necessary information to avoid false positives.

**Limitations:**

- The paper does not include a section on limitations/negative societal impact.
- An important challenge for segmentation in videos is temporary object disappearance due to occlusion, which results in false positives on a per-frame level. Since the semantic alignment ground truth is per-video rather than per-frame the method would not able to address this challenge, but if the ground truth were collected per-frame then perhaps it would.


**Strengths And Weaknesses:**

Strengths:

- The paper points out an important limitation of current R-VOS datasets and evaluation metrics and takes the proper steps to address it by creating the metric R and augmenting Ref-YouTube-VOS with negative video-text alignment pairs.
- The idea to enforce semantic consistency of the embedded textual space and generated embeddings is novel and interesting.
- The experiments are carefully designed. They clearly show that the model improves upon the state of the art. The ablation results show the benefit of the proposed relational cycle consistency losses.

Weaknesses:

- The paper is missing some details on the $R^{2}$-YouTube-VOS dataset including how many negative text-video pairs are created.

---

> ### Author Response · Authors · 2022-07-26
> **Response to Reviewer fwGH**
>
>
> We thank the reviewer for the particularly insightful feedback.
> The detailed comments have been incorporated in the revised version to improve the readability of the paper.
> Our answers to the questions are as follows.
>
> ---
> **1. Details in  $\textrm{R}^2$-Youtube-VOS dataset. How many negative text-video pairs are created?**
>
> We describe the $\textrm{R}^2$-Youtube-VOS dataset construction in Line 239-243. Here we supplement with details. The $\textrm{R}^2$-Youtube-VOS is an evaluation dataset that contains two parts, i.e., positive pairs and negative pairs. For positive pairs, the text-video pairs are exactly the same as the Ref-Youtube-VOS validation set. For negative pairs, the videos and texts are still selected from the Ref-Youtube-VOS validation set, while the order of videos is shuffled to make sure that each text queries to a different video. The number of negative pairs is the **same as positive pairs**. In this way, each text queries to **one** positive and **one** negative video.
>
> ---
> **2. In the sentence in lines 242-243, can you clarify what "constrain all negative text-video pair unrelated" means?**
>
> This sentence aims to emphasize that the text is ensured to query to a different video after shuffling the original video set. Thus, the text is not related to the queried video. We will clarify the expression in the revised paper.
>
> ---
> **3. Suggestions**
>
> Thanks for your constructive suggestions. We will revise the paper for the following items.
> We have revised the submission. The revised parts are marked in **blue for easy tracking. Here we list the changes.
>
> * We will add related reference on vision-language representation learning in Section 2.
>
> * We will correct the typos and adjust the unclear sentence according to the suggestions.
>
> * We summarize four differences between our work and ReferFormer, and will add the discussion in the revision.
> (1) Different from all the existing R-VOS methods, including ReferFormer, using all positive text-video pairs for training, we use both positive and negative pairs, which help the learning of differentiating semantic consensus between different pairs.
> (2) We leverage the relational text-video-text cycle consistency to better correspond the text embedding space to the video embedding space. Positive pairs are constrained with the cycle consistency for better embedding learning, while negative pairs unconstrained with the cycle constraint could be identified.
> (3) We utilize the early-grounding module, which modulate the video feature with the video-aware text embedding. Thus, irrelevant video features are suppressed in an early stage, while ReferFormer only uses dynamic convolution in the final mask decoding stage, easier to involve irrelevant objects, as shown in the results of positive pairs Figure 6.
> (4) Our instance query is composed of both the original sentence embedding and the reconstructed one. Different from ReferFormer that only utilizes original sentence embedding as queries, the reconstructed embedding can encode visual information to facilitate the instance query to decode the objects from visual features.
> We have already shown more qualitative results in the video demo of the supplemental material, and will add some samples in the main paper.
>
> ---
> **4. Limitation**
>
> We agree on the point that "per-frame false positive when object disappeared" can be a limitation of the current method. As the reviewer said, this is because the ground-truth label of RVOS is in the video-level, semantic alignment cannot be measured in a per-frame manner. We will add the limitation discussion to reveal this point for our future research.

---

> > ### Comment · Reviewer_fwGH · 2022-08-09
> > **Rebuttal response**
> >
> > Thank you for clarifying the differences between your work and Referformer. Points 3 and 4 are particularly useful in clarifying the difference in the method architecture which is most useful.
> >
> > The part of the rebuttal response to reviewer Gh1L, "As the positive video evaluation of $\mathcal{R}^{2}$-VOS is the same as Ref-Youtube-VOS, the numbers of J & F do not change [...]" would be useful to include in the main paper as well since this also clarifies how $\mathcal{R}^{2}$-VOS is different from Ref-Youtube-VOS.

---

> > > ### Author Response · Authors · 2022-08-09
> > > **Thank you for your suggestions!**
> > >
> > > Thank you for your further suggestions! We will revise the paper to reveal those changes.

---

### Author Response · Authors · 2022-08-07
**Thanks for all your comments and look forward to post-rebuttal feedbacks!**

Dear AC and all reviewers:

Thanks again for all of your constructive suggestions, which have helped us improve the quality and clarity of the paper!

Since the discussion phase is over half, we have not heard any post-rebuttal response yet.

Please don’t hesitate to let us know if there are any additional clarifications or experiments that we can offer, as we would love to introduce you to the merits of the paper. We appreciate your suggestions.

Best regards,

Authors of NeurIPS 2021 Submission #746

---

### Meta-Review · Area_Chair_LG46 · 2022-08-30

**Recommendation:** Reject
**Confidence:** Less certain

**Metareview:**

This paper presents an approach for video object segmentation. The paper considers the possibility that an (object) expression may not correspond to any object in the given video. The approach is based on relational cycle consistency, which the reviewers find technically sound. The paper also has a dataset contribution.

After the rebuttal and discussions, the reviewers still maintained split ratings. While two reviewers find the paper has its merit, Reviewer Fhve is against the paper, pointing out his/her concern regarding the experiments' fairness. Specifically, the reviewer points out that the comparison against the other works not using negative samples is unfair, as the training of the proposed approach benefits from such negative samples.

Although we agree to the authors' explanation that the ability to explicitly consider negative samples for out-of-distribution discrimination is the strength and that it is an interesting technical aspect of the paper, the paper lacks sufficient experiments to support the argument. It would have been better if the authors provided explicit experiments comparing their approach against the baselines like ReferFormer by adding classification heads for the background with negative samples in a meaningful way, as the authors also suggested. Also, there is a bit of novelty concern shared by Fhve and Gh1L.

Considering these aspects, the ACs recommend the rejection of the paper.

**Award:**

No

---

### Decision · Program_Chairs · 2022-09-14

Reject